# *In Vitro* Resistance against DNA Gyrase Inhibitor SPR719 in *Mycobacterium avium* and *Mycobacterium abscessus*

Wassihun Wedajo Aragaw,[a] Nicole Cotroneo,[b] Suzanne Stokes,[b] Michael Pucci,[b] Ian Critchley,[b] Martin Gengenbacher,[a,c] Thomas Dick[a,c,d]

[a]Center for Discovery and Innovation, Hackensack Meridian Health, Nutley, New Jersey, USA
[b]Spero Therapeutics, Cambridge, Massachusetts, USA
[c]Department of Medical Sciences, Hackensack Meridian School of Medicine, Nutley, New Jersey, USA
[d]Department of Microbiology and Immunology, Georgetown University, Washington, DC, USA

**ABSTRACT** The aminobenzimidazole SPR719 targets DNA gyrase in *Mycobacterium tuberculosis*. The molecule acts as inhibitor of the enzyme's ATPase located on the Gyrase B subunit of the tetrameric Gyrase $A_2B_2$ protein. SPR719 is also active against non-tuberculous mycobacteria (NTM) and recently entered clinical development for lung disease caused by these bacteria. Resistance against SPR719 in NTM has not been characterized. Here, we determined spontaneous *in vitro* resistance frequencies in single step resistance development studies, MICs of resistant strains, and resistance associated DNA sequence polymorphisms in two major NTM pathogens *Mycobacterium avium* and *Mycobacterium abscessus*. A low-frequency resistance ($10^{-8}$/CFU) was associated with missense mutations in the ATPase domain of the Gyrase B subunit in both bacteria, consistent with inhibition of DNA gyrase as the mechanism of action of SPR719 against NTM. For *M. abscessus*, but not for *M. avium*, a second, high-frequency ($10^{-6}$/CFU) resistance mechanism was observed. High-frequency SPR719 resistance was associated with frameshift mutations in the transcriptional repressor MAB_4384 previously shown to regulate expression of the drug efflux pump system MmpS5/MmpL5. Our results confirm DNA gyrase as target of SPR719 in NTM and reveal differential resistance development in the two NTM species, with *M. abscessus* displaying high-frequency indirect resistance possibly involving drug efflux.

**IMPORTANCE** Clinical emergence of resistance to new antibiotics affects their utility. Characterization of *in vitro* resistance is a first step in the profiling of resistance properties of novel drug candidates. Here, we characterized *in vitro* resistance against SPR719, a drug candidate for the treatment of lung disease caused by non-tuberculous mycobacteria (NTM). The identified resistance associated mutations and the observed differential resistance behavior of the two characterized NTM species provide a basis for follow-up studies of resistance *in vivo* to further inform clinical development of SPR719.

**KEYWORDS** DNA gyrase, GyrB, MAB_4384, NTM, non-tuberculous mycobacteria, SPR720, aminobenzimidazole

Address correspondence to Thomas Dick, thomas.dick.cdi@gmail.com.

The authors declare a conflict of interest. This investigation was in part funded by Spero Therapeutics. N.C., S.S., M.P., and I.C. are employees of Spero Therapeutics.

The prevalence of pulmonary disease caused by non-tuberculous mycobacteria (NTM) is increasing. The most frequently encountered species is the slow growing *Mycobacterium avium*, with high rates of the rapid growing *Mycobacterium abscessus* in certain geographic regions (1). Host risk factors include pre-existing lung conditions and immunomodulatory medications (1). Current multidrug regimens based on the macrolides clarithromycin or azithromycin require long duration of treatment that

**FIG 1** Structure of bioactive SPR719 and its phosphate prodrug SPR720.

often results in clinical failure (2). Thus, there is a medical need to develop new, well-tolerated, and more efficacious anti-NTM drugs (3).

The hetero-tetrameric DNA gyrase $GyrA_2B_2$ is a clinically validated target in myco-bacteria (4). The ATP-dependent type II topoisomerase is essential for reducing topo-logical strain caused by the unwinding of double-stranded DNA during transcription and replication. Furthermore, DNA gyrase is required for decatenation following repli-cation (4). The enzyme loops the DNA template(s) to form a crossing, then cuts one of the double helices and passes the other helix through the double strand broken DNA. Re-sealing the strand breaks completes the catalytic cycle. The process is driven by ATP hydrolysis catalyzed by the ATPase domain located in the N-terminal part of the Gyrase B subunits of the enzyme complex (4).

The DNA gyrase-targeting fluoroquinolone moxifloxacin is used for the treatment of macrolide resistant NTM lung disease. However, frequently encountered resistance limits the therapeutic utility of this drug class (5). Instead of classical enzyme inhibition, fluoroquinolones act as an "enzyme poison." The molecules bind to the breakage-reun-ion catalytic core of the enzyme composed of the N-terminal part of the Gyrase A and the C-terminal Toprim domain of the Gyrase B subunits, and trap the double strand broken DNA-protein complex (4).

The aminobenzamidazole SPR719 (VXc-486, Fig. 1) belongs to a novel class of DNA gyrase inhibitor with an on-target mechanism of action distinct from the fluoroquino-lones and not shared by currently marketed antibiotics. Targeting the ATPase domain of Gyrase B, the drug acts as competitive ATP binding inhibitor (6–9). This synthetic in-hibitor is active against *Mycobacterium tuberculosis* and non-tuberculous mycobacteria, including *M. avium* and *M. abscessus* (6, 10, 11). The phosphate prodrug of SPR719, SPR720 (Fig. 1), entered clinical development as a new oral option for treating myco-bacterial pulmonary infections (12). The drug candidate completed phase 1 studies to evaluate safety, tolerability, and pharmacokinetics (12).

Mechanism of action and *in vitro* development of resistance to SPR719 has been characterized in *M. tuberculosis* (6, 13). Spontaneous frequency of resistance was reported to be $\sim 10^{-8}$/CFU and resistance mutations mapped to its target, the ATPase domain of Gyrase B (6). Frequencies of resistance and mechanisms of resistance in NTM have not been determined and genetic evidence that anti-NTM activity is due to inhibition of Gyrase B is lacking. The objective of this study was to characterize *in vitro* resistance against SPR719 in the two major NTM pathogens *M. avium* and *M. abscessus* by determining frequencies of resistance, MICs of resistant strains, and resistance asso-ciated DNA sequence polymorphisms.

## RESULTS

To determine frequencies of resistance and isolate resistant mutants *in vitro* we used Middlebrook 7H10 agar containing bioactive SPR719 and *M. avium* subsp.

**TABLE 1** Characterization of SPR719-resistant *M. avium* subsp. *hominissuis* 109 mutants[a]

| Exp. | Strain | MIC (µM) | | | *gyrB* (nt/aa) | *gyrA* (nt/aa) |
| | | SPR719 | CLR | MXF | | |
|------|--------|---------|-----|-----|----------------|----------------|
| | wt | 6[b] | 2 | 6 | wt | wt |
| 1 | Spr^r1.1 | ~25[c] | 2 | 6 | T518C/I173T | wt |
| | Spr^r1.2 | ~25[c] | 2 | 6 | T518C/I173T | wt |
| 2 | Spr^r2.1 | ~25[c] | 2 | 6 | T518C/I173T | wt |
| | Spr^r2.2 | ~25[c] | 2 | 6 | T518C/I173T | wt |
| 3 | Spr^r3.1 | ~25[c] | 2 | 6 | T518C/I173T | wt |
| | Spr^r3.2 | ~25[c] | 2 | 6 | T518C/I173T | wt |

[a]Exp., independently grown culture batches; wt, wild type; Spr^r, SPR719-resistant strain; CLR, clarithromycin; MXF, moxifloxacin; nt/aa, nucleotide sequence polymorphism and associated amino acid substitution. The MIC experiments were carried out three times independently and the results are displayed as mean values.
[b]SPR719 concentration achieving 80% growth inhibition is given as the drug did not achieve 90% growth inhibition (see Fig. S1 for growth inhibition dose response curves).
[c]SPR719 MIC (concentration inhibiting 90% growth) was greater than 25 µM. However, at 25 µM, significant inhibition of growth was observed (~60%; see Fig. S1 for growth inhibition dose response curves). Concentrations were only tested up to 25 µM as at higher concentrations precipitation of the compound was observed.

*hominissuis* 109 and *M. abscessus* subsp. *abscessus* ATCC19977 as test strains (14, 15). First, pilot experiments were carried out to identify appropriate selection conditions. To determine the lowest drug concentrations that suppress outgrowth of wildtype colonies and thus allows selection of resistant strains, exponential phase culture samples from each species containing $\sim 2 \times 10^6$, $\sim 2 \times 10^7$ and $\sim 2 \times 10^8$ CFU were plated on agar containing 4, 8, and 16 x MIC SPR719, followed by incubation for 3 weeks (*M. avium*, MIC = 6 µM) or 1 week (*M. abscessus*, MIC = 1.6 µM). Colonies were restreaked on agar containing the same drug concentration as the agar on which they had emerged to determine their resistance status. Surprisingly, no colonies grew when *M. avium* was plated under these conditions. Thus, we repeated the experiment with agar containing 2 x MIC SPR719. This low drug concentration suppressed the outgrowth of wild type bacteria and resistant *M. avium* colonies emerged at a frequency of $10^{-8}$/ CFU. For *M. abscessus* the pilot experiments showed that agar containing 8 and 16 x MIC SPR719 prevented outgrowth of wild type colonies and indicated a frequency of resistance of $\sim 10^{-6}$/CFU.

After determining the selection conditions, resistant mutant isolation experiments were carried out three times with independently grown culture batches for each species using the identified selective drug concentrations to confirm frequencies of resistance and isolate resistant strains for phenotypic and genotypic characterization. Resistant mutant selection experiments were also carried out for the anti-NTM drug clarithromycin as comparator. Consistent with previous reports, frequency of resistance for the protein synthesis inhibitor was $\sim 10^{-8}$/CFU for both *M. avium* and *M. abscessus* (16, 17).

Plating of three independent cultures of *M. avium* on agar containing SPR719 at 2 x MIC confirmed a frequency of resistance of $\sim 10^{-8}$/CFU. The resistant colonies displayed homogenous wild type-like morphology. Two resistant strains from each selection experiment were randomly picked for further analyses. MIC determinations, carried out by the broth dilution method using 7H9 medium, showed a similar, moderate level of SPR719 resistance ($\sim$ 4-fold MIC increase) for all six strains (Table 1, Fig. S1). On drug-free agar, the mutants also grew like the wild type strain and (drug-free) growth curve measurements of a representative strain confirmed wild type-like growth in broth (Fig. S2), suggesting that the resistant mutations did not affect growth *in vitro*. Susceptibility to clarithromycin and moxifloxacin was not affected (Table 1), indicating that resistance is specific to SPR719 and not due to a multi-drug resistance mechanism. Targeted Sanger sequencing of the *M. avium gyrA* and *gyrB* genes revealed a single amino acid polymorphism, Ile173Thr, in the ATPase domain of Gyrase B in all six strains (Table 1). To determine whether the resistant strains harbor common additional, non-

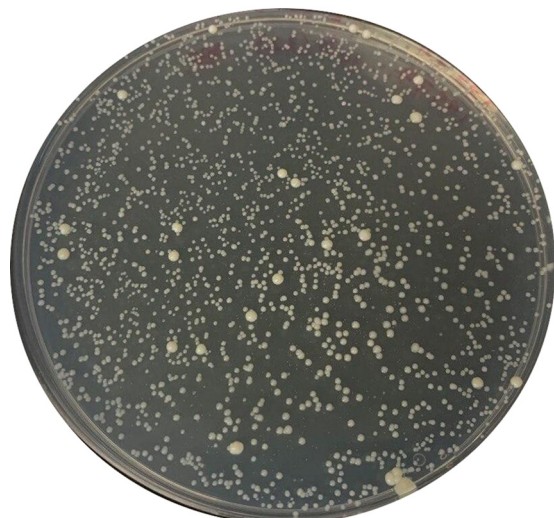

**FIG 2** Large and small colony morphotypes of SPR719-resistant *M. abscessus* subsp. *abscessus* ATCC 19977 on agar containing 16 x MIC of SPR719. $\sim 2 \times 10^9$ CFU of culture were plated and incubated for 1 week at 37°C.

*gyrB* polymorphisms, one strain from each selection experiment was subjected to whole genome sequencing. Polymorphisms in addition to the *gyrB* mutations were not observed. These results suggest that the Ile173Thr mutation in the ATPase domain of Gyrase B causes resistance to SPR719 in *M. avium*. It is interesting to note that an Ile-Thr substitution at this position in the amino acid sequence of Gyrase B was also reported to causes resistance against the DNA gyrase ATPase inhibitor coumermycin in *Staphylococcus aureus* (18). Taken together, analysis of resistance to SPR719 in *M. avium* revealed, similar to *M. tuberculosis* (6), a single low-frequency resistance mechanism involving missense mutations in the ATPase domain of Gyrase B. This result suggests that SPR719 targets DNA gyrase in *M. avium*.

Plating of three independent cultures of *M. abscessus* on agar media containing 8 and 16 x SPR719 MIC confirmed a frequency of resistance of $\sim 10^{-6}$/CFU at both drug concentrations. In contrast to the resistance experiments carried out for *M. avium*, resistant colonies in *M. abscessus* displayed two size-morphotypes emerging with different frequencies (Fig. 2). Wild type-sized large colonies emerged at a frequency of $\sim 10^{-8}$/CFU and small colonies emerged at a frequency of $\sim 10^{-6}$/CFU. Interestingly, the small to large colony size heterogeneity was only observed on agar containing 16 x MIC but not on agar containing 8 x MIC of SPR719. On drug-free agar, both morphotypes grew large, wild type-like colonies and growth curve determination of representative strains in (drug-free) broth showed wild type-like growth rates for both morphotypes (Fig. S3). Together, this suggests that the high-frequency mutations associated with the small colony phenotype observed on agar containing 16 x MIC SPR719 did not affect growth *per se* but was rather a drug dependent effect (i.e., that the small colony phenotype reflected a lower level of resistance; see below). Two large and two small colony resistant strains form each selection experiment were randomly picked for further analysis. MIC determinations for the six large colony strains showed similar, high-level SPR719 resistance ($>$ 16-fold MIC increase) for all strains (Table 2, Fig. S4). Susceptibility to clarithromycin and moxifloxacin was not affected (Table 2). Targeted Sanger sequencing of the *M. abscessus gyrA* and *gyrB* genes revealed a single amino acid polymorphism, Thr169Asn, in the ATPase domain of Gyrase B for all six strains (Table 2). To determine whether the resistant strains harbor common additional, non-*gyrB* DNA sequence polymorphisms, one large colony strain from each selection experiment was subjected to whole genome sequencing. Common polymorphisms in addition to the *gyrB* mutations were not observed (Table 2, Table S1). Together, these

**TABLE 2** Characterization of SPR719-resistant *M. abscessus* subsp. *abscessus* ATCC19977 mutants[a]

| Exp. | Strain | Colony size | MIC (µM) | | | *gyrB* (nt/aa) | *gyrA* (nt/aa) | Polymorphisms in MAB_4384 (InsDel) |
|---|---|---|---|---|---|---|---|---|
| | | | SPR719 | CLR | MXF | | | |
| 1 | wt | NA | 1.6 | 3 | 6 | wt | wt | wt |
| | Spr^r-L1.1 | L | >25[b] | 3 | 6 | C506A/T169N | wt | ND |
| | Spr^r-L1.2 | | >25[b] | 3 | 6 | C506A/T169N | wt | wt |
| | Spr^r-S1.1 | S | 12.5 | 3 | 6 | wt | wt | Ins247AC (fs) |
| | Spr^r-S1.2 | | 12.5 | 3 | 6 | wt | wt | Ins281A (sc) |
| 2 | Spr^r-L2.1 | L | >25[b] | 3 | 6 | C506A/T169N | wt | wt |
| | Spr^r-L2.2 | | >25[b] | 3 | 6 | C506A/T169N | wt | ND |
| | Spr^r-S2.1 | S | 12.5 | 3 | 6 | wt | wt | Del273_310 (fs) |
| | Spr^r-S2.2 | | 12.5 | 3 | 6 | wt | wt | Ins514A (fs) |
| 3 | Spr^r-L3.1 | L | >25[b] | 3 | 6 | C506A/T169N | wt | ND |
| | Spr^r-L3.2 | | >25[b] | 3 | 6 | C506A/T169N | wt | wt |
| | Spr^r-S3.1 | S | 12.5 | 3 | 6 | wt | wt | Ins31C (fs) |
| | Spr^r-S3.2 | | 12.5 | 3 | 6 | wt | wt | Ins31C (fs) |

[a]Exp., independently grown culture batches; wt, wild type; Spr^r, SPR719-resistant strain; L, S, large and small colony size phenotype observed on SPR719 containing agar; CLR, clarithromycin; MXF, moxifloxacin; nt/aa, nucleotide sequence polymorphism and associated amino acid substitution; InsDel, insertion or deletion resulting in frameshift (fs) or a stop codon (sc); NA, not applicable; ND, not determined. The MIC experiments were carried out three times independently and the results are displayed as mean values.

[b]>25, SPR719 MIC (concentration inhibiting 90% of growth) was greater than 25 µM. At 25 µM no significant inhibition of growth was observed (see Fig. S4 for growth inhibition dose response curves). Concentrations were only tested up to 25 µM as at higher concentrations precipitation of the compound was observed.

results suggest that the Thr169Asn mutation in the ATPase domain of Gyrase B causes resistance to SPR719 in *M. abscessus* and that the drug target in *M. abscessus* is DNA gyrase. These genetic results are consistent with recent biochemical analyses showing that SPR719 is a potent inhibitor of recombinant *M. abscessus* DNA gyrase (19). Thr169 in *M. abscessus* DNA gyrase corresponds to Ser169 in *M. tuberculosis* GyrB. Locher et al. (6) showed previously that mutations at this site in the gene of the tubercle bacillus (Ser169Ala) caused resistance to SPR719 by affecting binding of the drug. Interestingly, the same Ser169Ala mutation was also found to confer resistance against a different, pyrrolamide-based anti-tubercular Gyrase B ATPase inhibitor (20).

MIC determinations for the six small colony strains showed a lower level of SPR719 resistance with an 8-fold MIC increase (Table 2, Fig. S5). This lower resistance level appears to account for the small colony phenotype of these strains observed on agar containing 16 x MIC of SPR719 which was not observed on agar containing 8 x MIC SPR719 or under drug-free conditions. Susceptibility to clarithromycin or moxifloxacin was not affected (Table 2). Targeted sequencing of the *M. abscessus* gyrA and *gyrB* genes showed that the strains harbored wild type alleles for the gyrase encoding genes (Table 2). To identify the DNA sequence alterations associated with low-level resistance, the strains were subjected to whole genome sequencing, identifying several polymorphisms (Table S1). All six strains harbored distinct frameshift (i.e., likely loss-of-function) mutations in the transcriptional regulator MAB_4384 (Table 2) suggesting that these alterations are associated with the lower level SPR719 resistance phenotype. Mutations in this *M. abscessus* regulator were previously shown to cause resistance against thiacetazone analogs by allowing increased expression of the efflux pump system MmpS5/MmpL5 (21, 22). To determine whether the observed frameshift mutations in the regulator gene indeed cause resistance against SPR719 we complemented a representative low-level resistant strain with a wild type copy of MAB_4384 (Fig. 3). Introduction of a functional copy of the regulator into the resistant mutant background reverted the SPR719-resistant to a susceptible phenotype, suggesting that loss-of-function mutations in MAB_4384 are responsible for low-level SPR719 resistance (Fig. 3).

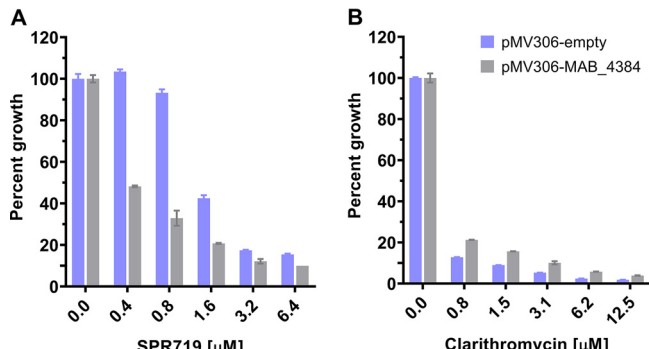

**FIG 3** Complementation of the low-level SPR719 resistance phenotype associated with frameshift mutations in *M. abscessus* ATCC 19977 MAB_4384. Plasmid pMV306 carrying a wild type copy of MAB_4384 (pMV306-MAB_4384) or not carrying any additional gene (pMV306-empty) was transformed into *M. abscessus* strain Spr$^r$-S2.1 harboring a frameshift mutation in MAB_4384 (Table 2) and growth inhibition dose response experiments with SPR719 (A) and clarithromycin as control (B) were carried out. (A) Introduction of MAB_4384-containing pMV306 into *M. abscessus* Spr$^r$-S2.1 resulted in increased susceptibility to SPR719. (B) Introduction of MAB_4384 containing pMV306 into *M. abscessus* Spr$^r$-S2.1 had no significant effect on susceptibility to clarithromycin. Experiments were carried out three times independently and mean values and standard deviations are shown.

## DISCUSSION

We show that spontaneous mutants of *M. avium* and *M. abscessus* displaying resistance to SPR719 occur at low-frequency in the ATPase domain of Gyrase B. This suggests DNA gyrase as the target of the drug candidate in NTM and is consistent with results previously generated for *M. tuberculosis* (6). Interestingly, Gyrase B mutant SPR719-resistant *M. avium* strains could only be isolated on agar containing 2 x MIC of SPR719. Higher drug concentrations did not yield any resistant colonies and thus appear to suppress the emergence of resistant strains. This is consistent with the finding that all resistant *M. avium* strains showed only a moderate $\sim$ 4-fold increase in MIC. In contrast, Gyrase B mutant SPR719-resistant *M. abscessus* strains could be isolated on agar containing 8 and 16 x MIC of SPR719. Accordingly, Gyrase B mutant *M. abscessus* displayed a strong, more than 16-fold increase in MIC. Consistent with the distinct on-target mechanisms of action of aminobenzimidazoles and fluoroquinolones, none of the SPR719-resistant *M. avium* and *M. abscessus* strains harboring Gyrase B mutations showed cross-resistance to moxifloxacin.

The reason for the intriguing difference in the resistance levels of GyrB-associated on-target mutations (Ile173Thr in *M. avium* vs Thr169Asn in *M. abscessus*) in the two mycobacterial species remains to be determined. To address these questions, detailed biochemical and structural studies of wild type and mutant DNA gyrases are required, which is beyond the scope of the current study. However, the finding that *M. avium* appears to develop only low-level resistance due to mutations in the DNA gyrase may be of relevance for the further clinical development of the drug candidate. If confirmed *in vivo* (in patients) this property would make the drug very useful for the treatment of *M. avium* infections as high-level, clinically significant resistance may only emerge at ultra-low frequencies.

For *M. abscessus*, high-frequency, lower level SPR719 resistance was observed, not reported for the *M. tuberculosis* (6), and not detected for *M. avium* in the present study. Lower level resistance was associated with frameshift mutations in the transcriptional regulator MAB_4384. Mutations in MAB_4384 were previously associated with a similar high frequency ($\sim 10^{-6}$/CFU) of resistance against thiacetazone analogs by Kremer and colleagues (21, 22). The authors showed that MAB_4384 acts as a repressor of the genes encoding the efflux pump system MmpS5/MmpL5. Mutations in MAB_4384 resulted in loss of repression and thus increased expression of the pump system and resistance against the thiacetazone analogs (21, 22). Thus, a similar efflux-based

mechanism may be the cause of MAB_4384 associated SPR719 resistance in *M. abscessus*. It is interesting to note that resistance to SPR719 in strains harboring MAB_4384 mutations is lower when compared with the resistance level conferred by the on-target mutation in *M. abscessus* GyrB. The reasons for this difference remain to be determined. However, there are other drugs showing higher resistance levels in on-target missense mutants (reducing/preventing binding) when compared with indirect resistance mechanisms such as pumps. A recent example is resistance in *M. tuberculosis* against the F-ATP synthase inhibitor bedaquiline. On-target mutations in the c subunit of the enzyme complex confer very high levels of resistance, whereas a pump-related resistance mechanism confers low levels of resistance (23).

One of the limitations of the current study was that the resistance associated polymorphisms in the *gyrB* genes of the two mycobacterial species were not genetically complemented with wild type alleles. Thus, formal proof that the observed *gyrB* polymorphisms are causal for resistance was not provided. However, a causal relationship is highly likely as amino acid substitutions at the same amino acid residue positions in the Gyrase B protein have been reported previously to cause resistance against ATPase inhibitors in other bacteria (6, 18).

The characterization of *in vitro* resistance described in the current work presents a first step in profiling of the resistance properties of this novel drug candidate for the treatment of NTM diseases. The identified resistance associated mutations and observed differential resistance behavior of the two characterized NTM species provides the basis for follow-up studies of resistance *in vivo* to guide the clinical development of SPR719. The relevance of the observed *in vitro* resistance mutations for the clinical use of SPR719/720 against NTM remains to be determined. It is important to consider that drugs for the treatment of NTM lung disease are not used as monotherapy, but in combination regimens (2), which should mitigate the risk of resistance development in patients.

## MATERIALS AND METHODS

**Strains, media, culture conditions, and drugs.** *M. avium* subsp. *hominissuis* 109 (14) was provided by Petros Karakousis (Johns Hopkins University). *M. abscessus* subsp. *abscessus* ATCC 19977 (15) was purchased from the American Type Culture Collection.

Strains were grown in complete Middlebrook 7H9 broth (BD Difco) supplemented with 0.05% Tween 80, 0.2% glycerol, and 10% albumin-dextrose-catalase with orbital shaking at 90 rpm (INFORS HT Multitron). For determination of CFU, bacterial cultures were spread onto Middlebrook 7H10 agar (BD Difco) supplemented with 10% (vol/vol) Middlebrook oleic acid-albumin-dextrose-catalase and 0.5% glycerol and grown at 37°C for 3 weeks (*M. avium*) or 1 week (*M. abscessus*). When appropriate, agar was supplemented with SPR719 or clarithromycin for isolation of resistant mutants. SPR719 was obtained from Spero Therapeutics. Clarithromycin and Moxifloxacin were purchased from Sigma-Aldrich, USA.

**Selection of mutants.** Mutant selection was carried out as described in Gopal et al. (24). In brief, bacterial inocula ($\sim$2 × 10$^6$, $\sim$2 × 10$^7$, and $\sim$2 × 10$^8$) from mid-log cultures of *M. avium* or *M. abscessus* were spread on 7H10 agar containing 2 x MIC of SPR719 (*M. avium*) or 8 and 16 x MIC (*M. abscessus*) and grown at 37°C for 3 weeks (*M. avium*) or 1 week (*M. abscessus*). To verify resistance, putative resistant colonies were picked and restreaked, alongside wild type bacteria, on agar containing the SPR719 concentrations on which the putative mutants were selected. This showed that all colonies isolated under these conditions were true resistant colonies. Single colonies were picked from the restreak plates, inoculated into 7H9 broth, expanded to mid-log phase, and stored with 10% glycerol at –80°C until used for subsequent studies.

**MIC measurement and determination of growth curves.** MIC was determined using the broth microdilution method in 7H9 as described previously with minor modifications (25). SPR719 and comparator antibiotics were serially diluted 2-fold in 100 $\mu$L volume of broth up to 10 concentration points in a 96-well plate format (flat bottom, Corning). Bacteria were grown to mid-log phase (OD$_{600}$ 0.4 to 0.6) and the cultures were diluted to OD$_{600}$ = 0.1. 100 $\mu$L of bacterial suspensions were dispensed into the wells, resulting in a final volume of 200 $\mu$L and a seeding density of OD$_{600}$ = 0.05. After reading at day 0, OD$_{600}$, plates were sealed with Nunc sealing tape (Thermo Scientific), placed in a humidified plastic box (Sterilite), and incubated for 4 days (*M. avium*) or 3 days (*M. abscessus*) with shaking at 90 rpm at 37°C. Growth inhibition was measured by reading optical density (OD$_{600}$) using TECAN infinite M200Pro microplate reader (TECAN). MIC values, defined as the concentration that inhibited 90% of bacterial growth compared with drug-free control unless stated otherwise, were deduced from the resulting dose response curves. Growth curves were determined in broth via OD$_{600}$ measurement of cultures growing in 1-L roller bottles (Corning) at 37°C (starting OD$_{600}$ = 0.05) using Ultrospec 10 cell density meter

(Biochrom, Holliston, MA, USA). GraphPad Prism 8 (GraphPad Software, Inc.) was used to determine the dose response and growth curves.

**DNA sequencing.** Genomic DNA was extracted using the phenol-chloroform method as described previously (26). Targeted sanger sequencing of *gyrA* and *gyrB* was performed by Genewiz Inc. (South Plainfield, NJ, USA) using primers listed in Table S2. Whole genome sequencing, including DNA quantification (Agarose gel electrophoresis and Qubit 3.0 fluorometer quantification), library construction (NEBNext DNA Library Prep Kit), high-throughput DNA sequencing (Illumina sequencing platform), and bioinformatic analyses (single nucleotide polymorphisms and insertions/deletions detection and annotation) were carried out by Novogene Corporation Inc. (Sacramento, CA, USA).

**Genetic complementation experiment.** *M. abscessus* ATTC 19977 Spr$^r$-S2.1, a representative low-level SPR719-resistant mutant harboring a frameshift mutation in MAB_4384 (Table 2), was complemented with a wild type copy of the MAB_4384 coding sequence expressed under the control of the hsp60 promoter as described previously (27). The respective 1,085-bp DNA fragment carrying the hsp60 promoter and the coding sequence was custom synthesized by Genewiz and inserted into the NotI-HindIII sites of plasmid pMV306 (28). Plasmids were electroporated into *M. abscessus* and transformants were selected on 400 $\mu$g/mL kanamycin agar as described previously (19). Dose response experiments were carried out as described under "MIC measurement and determination of growth curves."

**Data availability.** Whole-genome sequencing data of the large-colony mutants will be made available by the authors upon request.

## SUPPLEMENTAL MATERIAL

Supplemental material is available online only.
**SUPPLEMENTAL FILE 1**, PDF file, 0.4 MB.

## ACKNOWLEDGMENTS

We thank Petros Karakousis (Johns Hopkins University) for providing *M. avium* 109. The research reported in this publication is supported by the National Institute of Allergy and Infectious Diseases of the National Institutes of Health award number R01AI132374 and by Spero Therapeutics. The content is solely the responsibility of the authors and does not necessarily represent the official views of the National Institutes of Health.

Investigation: W.W.A.; Writing - original draft: W.W.A.; Writing - review & editing: all authors; Funding acquisition: T.D.; Supervision: N.C., M.G., and T.D.

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
