## [Reviewer comments · Microbiology Spectrum]

Microbiology Spectrum

In vitro resistance against DNA gyrase inhibitor SPR719 in *Mycobacterium avium* and *Mycobacterium abscessus*

Wassihun Aragaw, Nicole Cotroneo, Suzanne Stokes, Michael Pucci, Ian Critchley, Martin Gengenbacher, and Thomas Dick

Corresponding Author(s): Thomas Dick, Hackensack Meridian Health

Review Timeline:

Submission Date:	August 18, 2021
Editorial Decision:	September 16, 2021
Revision Received:	November 22, 2021
Editorial Decision:	December 7, 2021
Revision Received:	December 9, 2021
Accepted:	December 11, 2021

Editor: Amit Singh

Reviewer(s): Disclosure of reviewer identity is with reference to reviewer comments included in decision letter(s). The following individuals involved in review of your submission have agreed to reveal their identity: Vinayak Singh (Reviewer #1); Sheetal Gandotra (Reviewer #2)

Transaction Report:

DOI: <https://doi.org/10.1128/Spectrum.01321-21>

September 16, 2021

Prof. Thomas Dick
Hackensack Meridian Health
Center for Discovery and Innovation
111 Ideation Way
Building 102
Nutley, NJ 07110

Re: Spectrum01321-21 (In vitro resistance against DNA gyrase inhibitor SPR719 in *Mycobacterium avium* and *Mycobacterium abscessus*)

Dear Prof. Thomas Dick:

Thank you for giving us the opportunity to review your manuscript. The work is interesting and our reviewers are generally favorable. However, all of them pointed out that further consideration of this work requires genetic complementation of mutations. I agree with our reviewers and request you and coauthors to experimentally revise the manuscript in line with the comments.

Thank you for submitting your manuscript to Microbiology Spectrum. When submitting the revised version of your paper, please provide (1) point-by-point responses to the issues raised by the reviewers as file type "Response to Reviewers," not in your cover letter, and (2) a PDF file that indicates the changes from the original submission (by highlighting or underlining the changes) as file type "Marked Up Manuscript - For Review Only". Please use this link to submit your revised manuscript - we strongly recommend that you submit your paper within the next 60 days or reach out to me. Detailed information on submitting your revised paper are below.

Link Not Available

Sincerely,

Amit Singh

Journals Department
Reviewer comments:

Reviewer #1 (Comments for the Author):

Authors Aragaw et al. characterized the in vitro resistance of a known GyrB inhibitor SPR719 against the two non-tuberculous mycobacterial (NTM) pathogens *Mycobacterium avium* and *Mycobacterium abscessus* by raising the spontaneous resistant mutants which allowed them to determine the frequencies of resistance, MICs of resistant mutants, and mapping of mutations. While the MS reads very well it lacks in novelty (and in-depth investigation of the mechanism of resistance/action - such as validation of resistant mechanism by complementation of resistant mutants with wt copy of the *gyrB*, analysis of mutated GyrB vs wt GyrB for enzymatic activity and inhibition kinetics with SPR719, on-target activity using conditional knockdown etc. - most of these experiments are done by this lab in their recent publication) and for this reason, I find it very preliminary.

Reviewer #2 (Comments for the Author):

Aragaw et al identify frequency and likely cause of resistance against the DNA gyrase inhibitor SPR719 in two NTM species. This work is highly valuable for our understanding of emergence of drug resistance in pathogenic mycobacteria. The experimental approach is technically sound and the methods are described adequately. The authors find that in *M. avium*, a very low frequency resistant mutant emerges against SPR719, with moderate resistance while in *M. abscessus*, a higher frequency resistant mutant with stronger resistance emerges. While the authors describe mutations identified by sanger sequencing of *gyrB* and results of the WGS, their study does not adequately answer the findings of differences in degree of resistance.

Specific concerns:

1. The authors do not discuss the *GyrB* T518C mutation in the resistance clones of *M. avium* nor of *GyrB* C506A in resistant clones *M. abscessus*.
2. In the absence of functional data for any of these mutations on the resistance mechanism, it would be difficult to explain the differences in degree of resistance across the two species, or across the small and big colonies in case of *M. abscessus*.
3. What explains the relatively weaker resistance of the smaller colonies that harbour MAB-4384 mutations?

Reviewer #3 (Comments for the Author):

The present study by Aragaw et al titled "In vitro resistance against DNA gyrase inhibitor SPR719 in *Mycobacterium avium* and *Mycobacterium abscessus*" investigates in vitro resistance against a novel drug SPR719 in non-tuberculous mycobacteria.

The authors have identified and characterized spontaneous mutations in *M. avium* and *M. abscessus* that they concluded to be associated with SPR719 resistance. A missense mutation mapped to the ATPase domain of the Gyrase B subunit in both bacteria. It is interesting to note the second high frequency resistance mechanism involving regulation of drug efflux system in *M. abscessus*. To sum up, the study throws light into the possible mechanisms of resistance against SPR719 in two NTM strains that could be useful in treatment of NTMs.

General Comments:

Overall the manuscript reads well. However, at some points the text appears repetitive and thus writing could be improved in some sections.

Some supplementary figures could be moved to the main body to support the Table data. It will be easier for the reader to follow the data if some of the growth curves were included as main figures

Specific Comments:

1. There is insufficient proof that the spontaneous mutations identified in the study are directly responsible for the observed resistance against SPR719.
2. No data is provided regarding the effect of these mutations and SPR719 resistance in vivo. To better understand the resistance mechanisms under clinical settings, the authors should consider experimental approaches in vivo or using ex vivo cell culture models to help support the conclusions.
3. Statistical analysis is missing.

Staff Comments:

Preparing Revision Guidelines

- Point-by-point responses to the issues raised by the reviewers in a file named "Response to Reviewers," NOT IN YOUR COVER LETTER.
- Upload a compare copy of the manuscript (without figures) as a "Marked-Up Manuscript" file.

- Each figure must be uploaded as a separate file, and any multipanel figures must be assembled into one file.
- Manuscript: A .DOC version of the revised manuscript
- Figures: Editable, high-resolution, individual figure files are required at revision, TIFF or EPS files are preferred

Please return the manuscript within 60 days; if you cannot complete the modification within this time period, please contact me. If you do not wish to modify the manuscript and prefer to submit it to another journal, please notify me of your decision immediately so that the manuscript may be formally withdrawn from consideration by Microbiology Spectrum.

The present study by Aragaw et al titled “In vitro resistance against DNA gyrase inhibitor SPR719 in *Mycobacterium avium* and *Mycobacterium abscessus*” investigates in vitro resistance against a novel drug SPR719 in non-tuberculous mycobacteria.

The authors have identified and characterized spontaneous mutations in *M. avium* and *M. abscessus* that they concluded to be associated with SPR719 resistance. A missense mutation mapped to the ATPase domain of the Gyrase B subunit in both bacteria. It is interesting to note the second high frequency resistance mechanism involving regulation of drug efflux system in *M. abscessus*. To sum up, the study throws light into the possible mechanisms of resistance against SPR719 in two NTM strains, that could be useful in treatment of NTMs.

General Comments: Overall the manuscript reads well. However, at some points the text appears repetitive and thus writing could be improved in some sections.

Some supplementary figures could be moved to the main body to support the Table data. It will be easier for the reader to follow the data if some of the growth curves were included as main figures.

Specific Comments:

1. There is insufficient proof that the spontaneous mutations identified in the study are directly responsible for the observed resistance against SPR719.
2. No data is provided regarding the effect of these mutations and SPR719 resistance in vivo. To better understand the resistance mechanisms under clinical settings, the authors should consider experimental approaches in vivo or using ex vivo cell culture models to help support the conclusions.
3. Statistical analysis is missing.

Response to editor and reviewers: Spectrum01321-21 (In vitro resistance against DNA gyrase inhibitor SPR719 in Mycobacterium avium and Mycobacterium abscessus)

Editor's comment (16 Sept 21)

The work is interesting and our reviewers are generally favorable. However, all of them pointed out that further consideration of this work requires genetic complementation of mutations. I agree with our reviewers and request you and coauthors to experimentally revise the manuscript in line with the comments.

Authors clarification to the editor (17 Sept 21)

Regarding the point that the reviewers raised: complementation of the resistant strains to confirm that the observed polymorphisms indeed cause the observed phenotype. This is a good point. In fact, we discuss this point rather extensively in our discussion as a limitation. We also discuss why we think that this limitation is acceptable. Regarding the mutations in the gyrase genes of M. abscessus and M. avium: the mutant strains were isolated in independent selection experiments; and the same (identical) amino acid substitutions have been described to cause resistance against ATPase inhibitors previously. Thus, it is highly plausible that these mutations are causal for resistance. A similar rationale can be applied to the indirect resistance mechanism observed for M. abscessus. FYI: We just published earlier this year (as an example) a manuscript using the same rationale – and that was acceptable by AAC standards (Ganapathy et al. (2021), AAC <https://doi.org/10.1128/AAC.02420-20>). Furthermore, I would like to note that the focus of this work on the clinical candidate SPR719 was on the determination of the frequencies of resistance, characterization of the MIC levels of resistant mutant classes and their associated polymorphisms (as stated in the manuscript) – i.e., not on mechanisms per se. The former data are a requirement by the FDA: Resistance frequencies and resulting MIC levels present critical information for further clinical development. Thus, the current work presents important (clinically relevant) information on this new antibiotic candidate and thus needs to get into the public domain. In summary, we agree that not having done genetic complementation experiments presents a limitation of the work (and we state this clearly). At the same time, we think the plausibility argument is reasonable. Finally, there are ample examples in the literature where resistance analyses are published without complementation, showing that this strategy is acceptable (under the condition that the data situation and rationale is similar to the current manuscript). In light of this additional background information and explanations, I would like to ask whether you still require formal genetic complementation experiments as a condition for accepting our manuscript; or whether you would reconsider your decision.

Editor's comment on authors' clarification (20 Sept 21)

Thanks for your email and your response clarifying reviewers' comments. I broadly agree with your clarifications. Our reviewers are also convinced about the gyrase B subunit mutations as these have been previously reported in other bacteria. However, polymorphism in the transcriptional repressor is unique and requires further characterization. Our decision is largely based on the reviewers' comments and in this case all three reviewers unanimously raised similar concerns. My suggestion is to address the reviewers' concerns as much as possible

and/or provide an in-depth discussion or any new analysis on why these mutations are likely responsible for resistance in the response letter and in the revised manuscript.

Authors: Thank you for your understanding and thank you for agreeing that confirmation of the resistance mechanism due to mutations in gyrase B is not essential. We carried now out complementation for the loss of function mutations in the transcriptional repressor as requested and added the data to the manuscript. We also added the corresponding materials and methods.

Reviewer comments:

Reviewer #1 (Comments for the Author):

Authors Aragaw et al. characterized the in vitro resistance of a known GyrB inhibitor SPR719 against the two non-tuberculous mycobacterial (NTM) pathogens Mycobacterium avium and Mycobacterium abscessus by raising the spontaneous resistant mutants which allowed them to determine the frequencies of resistance, MICs of resistant mutants, and mapping of mutations. While the MS reads very well it lacks in novelty (and in-depth investigation of the mechanism of resistance/action - such as validation of resistant mechanism by complementation of resistant mutants with wt copy of the gyrB, analysis of mutated GyrB vs wt GyrB for enzymatic activity and inhibition kinetics with SPR719, on-target activity using conditional knockdown etc. - most of these experiments are done by this lab in their recent publication) and for this reason, I find it very preliminary.

Authors: thank you for your comments. Regarding ‘the work lacks novelty and in-depth investigation’. We don’t agree with the statement that the work ‘lacks novelty’ but we do agree that this work does not present an ‘in-depth investigation’ (of mechanisms). Characterization of in vitro resistance in NTM against the clinical candidate SPR719/720 has not been reported. Thus, this work contains novel data. These novel data are highly relevant to the further clinical development of this candidate (for instance one may favor to develop the drug for M. avium first). Indeed, this work does not describe an in-depth investigation of mechanisms. But this was not the objective of the current study. For further clinical development it is important to determine frequencies of resistance, classes of resistance mutants, possible in vitro fitness effects of resistance mutations, and the resistance associated DNA polymorphisms. That was the objective of the current study (as stated in the abstract and in the introduction; see label in marked-up manuscript) and these data were delivered.

The gyrase B mutations we find associated with SPR719 resistance have been identified in other bacteria to cause resistance against ATPase inhibitors. Thus, it is highly likely, that these mutations are also causal for resistance against NTM. We discuss this point extensively in the discussion and also clearly state that the lack of complementation presents as a limitation of the study (see label in the discussion of the marked-up manuscript).

Taken together, we would like to emphasize that this study is not about in-depth analyses of mechanisms but a standard (and rather comprehensive) in vitro resistance analysis supporting and guiding clinical development of a drug candidate.

Reviewer #2 (Comments for the Author):

Aragaw et al identify frequency and likely cause of resistance against the DNA gyrase inhibitor SPR719 in two NTM species. This work is highly valuable for our understanding of emergence of drug resistance in pathogenic mycobacteria. The experimental approach is technically sound and the methods are described adequately. The authors find that in *M. avium*, a very low frequency resistant mutant emerges against SPR719, with moderate resistance while in *M. abscessus*, a higher frequency resistant mutant with stronger resistance emerges. While the authors describe mutations identified by sanger sequencing of *gyrB* and results of the WGS, their study does not adequately answer the findings of differences in degree of resistance.

Thank you for your comments and positive evaluation.

Specific concerns:

1. The authors do not discuss the GyrB T518C mutation in the resistance clones of *M. avium* nor of GyrB C506A in resistant clones *M. abscessus*.

Authors: That is correct. We do not discuss (address the possible reasons for) why we are getting two distinct missense mutations and why the respective resistance levels differ. The differences in the resistance levels associated with the (different) GyraseB missense mutations in the different mycobacterial species is indeed intriguing. We state the difference in our manuscript, and also say that the reason for this difference remains to be determined (see label in results of the marked-up manuscript). The reason we don't discuss this difference in more detail is because we don't have anything reasonable to say or to speculate. The finding is 'puzzling' and requires detailed structural (Xray structures of wild types and mutants, with and without bound inhibitor) and biochemical analyses of the two DNA gyrases. We think that this detailed analysis is beyond the scope of the current work. We would like to note that our finding that *M. avium* appears to develop only low level resistance due to mutations in the DNA gyrase is of high relevance for the further clinical development of the drug candidate. If confirmed in vivo (in patients) this property would make the drug very useful for the treatment of *M. avium* infections (high level, clinically significant resistance may only emerge at ultra low frequencies).

2. In the absence of functional data for any of these mutations on the resistance mechanism, it would be difficult to explain the differences in degree of resistance across the two species, or across the small and big colonies in case of *M. abscessus*.

Authors: Agree. Detailed structural, biochemical (and genetic) analyses would be required to address these questions. Again, as this study is focused on in vitro NTM resistance characterization to support further clinical development, we feel that these additional works are beyond the scope of the current manuscript.

3. What explains the relatively weaker resistance of the smaller colonies that harbour MAB-4384 mutations?

Authors: Good question – and we don't know. Detailed mechanistic studies are required to answer this question. In our experience in the lab, we typically observe higher resistance levels in on-target missense mutants (reducing/preventing binding) when compared to indirect resistance mechanisms (such as pumps). This may be similar here. Without any supporting data we prefer not to speculate.

Reviewer #3 (Comments for the Author):

The present study by Aragaw et al titled "In vitro resistance against DNA gyrase inhibitor SPR719 in *Mycobacterium avium* and *Mycobacterium abscessus*" investigates in vitro resistance against a novel drug SPR719 in non-tuberculous mycobacteria.

The authors have identified and characterized spontaneous mutations in *M. avium* and *M. abscessus* that they concluded to be associated with SPR719 resistance. A missense mutation mapped to the ATPase domain of the Gyrase B subunit in both bacteria. It is interesting to note the second high frequency resistance mechanism involving regulation of drug efflux system in *M. abscessus*. To sum up, the study throws light into the possible mechanisms of resistance against SPR719 in two NTM strains that could be useful in treatment of NTMs.

General Comments:

Overall the manuscript reads well. However, at some points the text appears repetitive and thus writing could be improved in some sections.

Authors: Thank you for your comment. We carried out two projects – resistance characterization in *M. avium* and in *M. abscessus*. We followed the same general procedure twice. This may explain the impression that the text feels somewhat repetitive. However, the results we obtained for the two mycobacterial species when we executed the two resistance characterization analyses were very different. This, we feel, required a detailed description of the process and the results. We also wanted to ensure that our work can be reproduced, again requiring detailed description of process and results. This again, may contribute to the impression of being somewhat repetitive. We would prefer not to shorten the write up as we think this would affect the ability of other groups to repeat our work. Of note: many published resistance mutant selection experiments are described very briefly which makes interpretation and reproducibility of the data difficult. We wanted to avoid this.

Some supplementary figures could be moved to the main body to support the Table data. It will be easier for the reader to follow the data if some of the growth curves were included as main figures

Authors: Thank you for this suggestion. We think that showing the real data of 'MIC experiments' (i.e. the dose response curves) is indeed important. Although the MIC values derived from these curves are reported in the tables of the main manuscript, the interested expert in the field may want to have close look at the actual curves. That's why we would like to publish not only the MIC values in Tables of the main manuscript (what most authors do) but also the corresponding dose response curves. However, putting 21 (!) dose response curves into the main manuscript may be a little of an overkill. After all,

these are ‘only’ standard dose response MIC curves and the key data points, the MIC values derived from these curves, are described in the Tables. Taken together, and considering the redundancy of information, we find it more appropriate to show these data for reference in the supplemental materials. We hope this makes sense and is acceptable. We did put the new data figure (Fig. 3) showing the results of the complementation experiment of the MAB_4384 mutant in the main text.

Specific Comments:

1. There is insufficient proof that the spontaneous mutations identified in the study are directly responsible for the observed resistance against SPR719.

Authors: Please see discussion with the editor above; here re-inserted.

Regarding the point that the reviewers raised: complementation of the resistant strains to confirm that the observed polymorphisms indeed cause the observed phenotype. This is a good point. In fact, we discuss this point rather extensively in our discussion as a limitation. We also discuss why we think that this limitation is acceptable. Regarding the mutations in the gyrase genes of *M. abscessus* and *M. avium*: the mutant strains were isolated in independent selection experiments; and the same (identical) amino acid substitutions have been described to cause resistance against ATPase inhibitors previously. Thus, it is highly plausible that these mutations are causal for resistance. A similar rationale can be applied to the indirect resistance mechanism observed for *M. abscessus*. FYI: We just published earlier this year (as an example) a manuscript using the same rationale – and that was acceptable by AAC standards (Ganapathy et al. (2021), AAC <https://doi.org/10.1128/AAC.02420-20>). Furthermore, I would like to note that the focus of this work on the clinical candidate SPR719 was on the determination of the frequencies of resistance, characterization of the MIC levels of resistant mutant classes and their associated polymorphisms (as stated in the manuscript) – i.e., not on mechanisms per se. The former data are a requirement from the FDA: Resistance frequencies and resulting MIC levels present critical information for further clinical development. Thus, the current work presents important (clinically relevant) information on this new antibiotic candidate and thus needs to get into the public domain. In summary, we agree that not having done genetic complementation experiments presents a limitation of the work (and we state this clearly). At the same time, we think the plausibility argument is reasonable. Finally, there are ample examples in the literature where resistance analyses are published without complementation, showing that this strategy is acceptable (under the condition that the data situation and rationale is similar to the current manuscript). In light of this additional background information and explanations, I would like to ask whether you still require formal genetic complementation experiments as a condition for accepting our manuscript; or whether you would reconsider your decision.

Editor’s comment on authors’ clarification (20 Sept 21)

Thanks for your email and your response clarifying reviewers' comments. I broadly agree with your clarifications. Our reviewers are also convinced about the gyrase B subunit mutations as these have been previously reported in other bacteria. However, polymorphism in the

transcriptional repressor is unique and requires further characterization. Our decision is largely based on the reviewers' comments and in this case all three reviewers unanimously raised similar concerns. My suggestion is to address the reviewers' concerns as much as possible and/or provide an in-depth discussion or any new analysis on why these mutations are likely responsible for resistance in the response letter and in the revised manuscript.

Authors: Thank you for agreeing that confirmation of the resistance mechanism due to mutations in gyrase B is not essential.

We carried now out complementation for the loss of function mutations in the transcriptional repressor and added the data to the manuscript (please see labelled part in results section and new Figure 3). We also added the additional materials and methods. The results show that adding a wild type copy of MAB_4384 into MAB_4384-mutant background reverses the resistance phenotype. This suggests that the polymorphisms observed in MAB_4384 are indeed causing the SPR719 resistance phenotype.

2. No data is provided regarding the effect of these mutations and SPR719 resistance in vivo. To better understand the resistance mechanisms under clinical settings, the authors should consider experimental approaches in vivo or using ex vivo cell culture models to help support the conclusions.

Authors: That is a very good point. The characterization of in vitro resistance is only a first step (as we state in the manuscript). We feel that isolation and characterization of in vivo mutants would be beyond the scope of the current work. Importantly, resistance will be monitored in the clinical trials in patients. Our in vitro resistance data provide a valuable baseline for the study of clinical (in vivo) resistance. Please note that we mention the limitation of the in vitro work in the manuscript. We state that the relevance of the polymorphisms identified in vitro for the clinical use remains to be determined (see label in discussion of the marked-up manuscript).

3. Statistical analysis is missing.

Authors: We carried out a number of dose response determinations and (drug free) growth curve measurements. All these experiments were carried out three times independently and mean values with standard deviations are shown. This is stated in the legends. (see labels in Table legends of marked-up manuscript and the Figure legends of the marked-up supplemental materials).

We would like to thank again the editor and the three reviewers for their thoughts and comments. We hope that the clarifications we provide (and the additional data set) are useful and make sense. Again, this is not about the in-depth mechanistic characterization of the mechanisms of action and resistance of a known GyraseB inhibitor in NTM (as interesting as this may be). This work provides critical in vitro resistance patterns and properties of a drug that is in clinical development against NTM.

December 7, 2021

Prof. Thomas Dick
Hackensack Meridian Health
Center for Discovery and Innovation
111 Ideation Way
Building 102
Nutley, NJ 07110

Re: Spectrum01321-21R1 (In vitro resistance against DNA gyrase inhibitor SPR719 in *Mycobacterium avium* and *Mycobacterium abscessus*)

Dear Prof. Thomas Dick:

Our reviewers are mostly satisfied with the revision and I am willing to accept the manuscript pending minor changes. As suggested by one of our reviewer: (i) Please include reasons for differences in resistance mechanisms between the two NTM species studied in the discussion section.

Data in Figure 3 should also include comparison with the wild type strain. If you have this data, please include in figure 3.

Thank you for submitting your manuscript to Microbiology Spectrum. As you will see your paper is very close to acceptance. Please modify the manuscript along the lines I have recommended. As these revisions are quite minor, I expect that you should be able to turn in the revised paper in less than 30 days, if not sooner. If your manuscript was reviewed, you will find the reviewers' comments below.

When submitting the revised version of your paper, please provide (1) point-by-point responses to the issues I raised in your cover letter, and (2) a PDF file that indicates the changes from the original submission (by highlighting or underlining the changes) as file type "Marked Up Manuscript - For Review Only". Please use this link to submit your revised manuscript. Detailed instructions on submitting your revised paper are below.

Link Not Available

Sincerely,

Amit Singh

Reviewer comments:

Reviewer #2 (Comments for the Author):

My questions are answered by the authors adequately in the response letter. Some of the points made by the authors in the response letter pertaining to reasons for differences in resistance mechanisms between the two NTM species studied should find a place in the discussion section, indeed being pointed out as speculative.
Data in Figure 3 should also include comparison with the wild type strain.

Preparing Revision Guidelines

- point-by-point responses to the issues I raised in your cover letter
- Upload a compare copy of the manuscript (without figures) as a "Marked-Up Manuscript" file.
- Each figure must be uploaded as a separate file, and any multipanel figures must be assembled into one file.
- Manuscript: A .DOC version of the revised manuscript
- Figures: Editable, high-resolution, individual figure files are required at revision, TIFF or EPS files are preferred

Please return the manuscript within 60 days; if you cannot complete the modification within this time period, please contact me. If you do not wish to modify the manuscript and prefer to submit it to another journal, please notify me of your decision immediately so that the manuscript may be formally withdrawn from consideration by Microbiology Spectrum.

Nutley, 9 December 2021

Dear Dr. Singh,

Re: revision of revised manuscript Spectrum01321-21R1 (In vitro resistance against DNA gyrase inhibitor SPR719 in *Mycobacterium avium* and *Mycobacterium abscessus*)

Please find enclosed our revision of our revised manuscript. As requested I include our response to your comments and the comments from reviewer#2 in this cover letter.

Reviewer #2:

-Some of the points made by the authors in the response letter pertaining to reasons for differences in resistance mechanisms between the two NTM species studied should find a place in the discussion section, indeed being pointed out as speculative.

-Data in Figure 3 should also include comparison with the wild type strain.

Editor:

(i) Please include reasons for differences in resistance mechanisms between the two NTM species studied in the discussion section.

(ii) Data in Figure 3 should also include comparison with the wild type strain. If you have this data, please include in figure 3.

Regarding (i)

We extend now the discussion on the differences in resistance between the two NTM species in the discussion section by transferring comments from the response letter to the discussion as requested by reviewer#2 (see Marked-up manuscript). The changes are as follows:

Addressing the following in our response letter

“1. The authors do not discuss the GyrB T518C mutation in the resistance clones of *M. avium* nor of GyrB C506A in resistant clones *M. abscessus*.

Authors: That is correct. We do not discuss (address the possible reasons for) why we are getting two distinct missense mutations and why the respective resistance levels differ. The differences in the resistance levels associated with the (different) GyraseB missense mutations in the different mycobacterial species is indeed intriguing. We state the difference in our manuscript, and also say that the reason for this difference remains to be determined (see label in results of the marked-up manuscript). The reason we don't discuss this difference in more detail is because we don't have anything reasonable to say or to speculate. The finding is 'puzzling' and requires detailed structural (Xray structures of wild types and mutants, with and without bound inhibitor) and biochemical analyses of the two DNA gyrases. We think that this detailed analysis is beyond the scope of the current work. We would like to note that our finding that *M. avium* appears to develop only low level resistance due to mutations in the DNA gyrase is of high relevance for the further clinical development of the drug candidate. If confirmed in vivo (in patients) this property would make the drug very useful for the treatment of *M. avium* infections (high level, clinically significant resistance may only emerge at ultra low frequencies).”

Accordingly, we added now the following to the discussion:

Member of Hackensack Meridian Health

“The reason for the intriguing difference in the resistance levels of GyrB-associated on-target mutations (Ile173Thr in *M. avium* vs Thr169Asn in *M. abscessus*) in the two mycobacterial species remains to be determined. To address these questions, detailed biochemical and structural studies of wild type and mutant DNA gyrases are required, which is beyond the scope of the current study. However, the finding that *M. avium* appears to develop only low level resistance due to mutations in the DNA gyrase may be of relevance for the further clinical development of the drug candidate. If confirmed *in vivo* (in patients) this property would make the drug very useful for the treatment of *M. avium* infections as high level, clinically significant resistance may only emerge at ultra low frequencies.”

Addressing the following in our response letter

“3. What explains the relatively weaker resistance of the smaller colonies that harbour MAB-4384 mutations?”

Authors: Good question – and we don’t know. Detailed mechanistic studies are required to answer this question. In our experience in the lab, we typically observe higher resistance levels in on-target missense mutants (reducing/preventing binding) when compared to indirect resistance mechanisms (such as pumps). This may be similar here. Without any supporting data we prefer not to speculate.”

Accordingly, we added now the following to the discussion:

“It is interesting to note that resistance to SPR719 in strains harboring MAB_4384 mutations is lower when compared to the resistance level conferred by the on-target mutation in *M. abscessus* GyrB. The reasons for this difference remain to be determined. However, there are other drugs showing higher resistance levels in on-target missense mutants (reducing / preventing binding) when compared to indirect resistance mechanisms such as pumps. A recent example is resistance in *M. tuberculosis* against the F-ATP synthase inhibitor bedaquiline. On-target mutations in the c subunit of the enzyme complex confer very high levels of resistance, whereas a pump-related resistance mechanism confers low level of resistance (Andries K, et al., 2014. Acquired resistance of Mycobacterium tuberculosis to bedaquiline. PLoS one 9:e102135-e102135).”

Regarding (ii)

I am not sure the strain ATCC19977 without empty plasmid is required for Fig. 3. The important comparison is between the strain carrying the empty plasmid (control) and the strain carrying the plasmid with inserted MAB_4384 gene. Does adding a wild type copy of MAB_4384 affect the MIC? Thus, we did not include the strain without the empty plasmid when we carried out the experiment shown in Fig. 3; i.e. we don’t have this data set and thus can’t add it. The editor appears to be ok with that: “If you have this data, please include in figure 3.” Well, we don’t have the data. However, please note: of course we compared the MIC curve of the strain with empty plasmid with the MIC curves of the strain without empty plasmid obtained from prior experiments (to make sure that everything is consistent and that the empty plasmid does not affect the MIC curve) and find that the curves are -as expected- similar. In fact, the MIC curve of the ATCC19977 strain (without empty plasmid) can be found in the supplemental materials (Fig. S4, S5) where we included the strain as comparator for the MIC curves obtained for our SPR719-resistant ATCC19977 strains.

Taken together, we addressed both issues raised. We included a discussion on the differences in the resistance in the discussion, and we clarified the issue around including the ATCC19977 strain (without empty plasmid) in Fig. 3 (we didn’t run the strain without empty plasmid when we carried out the experiment shown in Fig. 3 and thus can’t include the data; however we did run MIC curves for the ATCC19977 strain before in other experiments [and the data are shown in the supplemental materials] and the MIC curves are similar as expected).

Member of Hackensack Meridian *Health*

111 Ideation Way, Nutley, NJ 07110

Sincerely,

Thomas Dick, PhD
Professor & Member
Phone: 201.880.3530
Email: thomas.dick.cdi@gmail.com
<https://hmh-cdi.org/our-team/dick-lab/>

December 11, 2021

Prof. Thomas Dick
Hackensack Meridian Health
Center for Discovery and Innovation
111 Ideation Way
Building 102
Nutley, NJ 07110

Re: Spectrum01321-21R2 (In vitro resistance against DNA gyrase inhibitor SPR719 in Mycobacterium avium and Mycobacterium abscessus)

Dear Prof. Thomas Dick:

Your manuscript has been accepted, and I am forwarding it to the ASM Journals Department for publication. You will be notified when your proofs are ready to be viewed.

Authors are requested to include a Data Availability statement for the whole genome sequence(s) of the large colony from each selection experiment. This can be done during the proof reading stage.

Sincerely,

Amit Singh
Editor, Microbiology Spectrum

Journals Department
Supplemental file 1: Accept